# Characterization of the TREM-1 signaling landscape in human neutrophils

**Frederic Ries[1], Matthias Klein[2,3], Nora Rogmann[1], Sophie Többen[1], Federico Marini[3,4], Florian Heidel[5,6], Markus P. Radsak[1,3¤]***

**1** Department of Hematology and Medical Oncology, University Medical Center of the Johannes Gutenberg-University, Mainz, Germany, **2** Institute for Immunology, University Medical Center of the Johannes Gutenberg-University, Mainz, Germany, **3** Research Center for Immunotherapy (FZI), Mainz, Germany, **4** Institute of Medical Biostatistics, Epidemiology and Informatics (IMBEI), University Medical Center of the Johannes Gutenberg-University, Mainz, Germany, **5** Department of Hematology, Hemostasis, Oncology and Stem Cell Transplantation, Hannover Medical School (MHH), Hannover, Germany, **6** Leibniz-Institute on Aging, Fritz-Lipmann-Institute, Jena, Germany

¤ Current address: Department of Oncology and Hematology, Donau-Isar-Klinikum, Deggendorf, Germany
* radsak@uni-mainz.de

## Abstract

The Triggering Receptor Expressed on Myeloid Cells (TREM)-1 is a member of the Immunoglobulin superfamily, and an activating receptor mainly expressed on myeloid cells. Beyond its role in acute and chronic inflammatory processes, TREM-1 is also involved in cancer emergence and progression probably by alteration of the tumor-associated neutrophils (TAN) and macrophages (TAM). Advanced information about the TREM-1 signaling cascade may reveal novel targets for treating inflammatory and cancer diseases. As many specific kinase inhibitors are approved for treating various diseases, targeting kinases being active after TREM-1 ligation serves as a promising approach. Therefore, we investigated the protein tyrosine kinome (PTK) and serine threonine kinome (STK) by kinome activity profiling of purified human neutrophils after TREM-1 activation. As TREM-1 interacts with Toll-like receptor (TLR) 4 signaling, we used TLR4-activation by LPS to define TREM-1 specific pathways. We found an increased kinome activity after receptor ligation and could predict individual kinases. To gain further insights into the signaling cascade, we additionally investigated the transcriptomic profile that made us available to link the kinome activity to the resulting transcriptomic profile. In sum, we revealed several signaling pathways being active after TREM-1 ligation that are associated with various biological processed and diseases. This study facilitates selecting kinase inhibitors for further validation with the aim of targeting TREM-1 signaling in various inflammatory or cancer disease conditions.

**Data availability statement:** All relevant data for this study are publicly available from the NCBI GEO repository (https://www.ncbi.nlm.nih.gov/geo/query/acc.cgi?acc=GSE286969).

**Funding:** This work was supported by the Deutsche Forschungsgemeinschaft, (German Research Foundation): grant CRC1066 TP B18 (MPR), German Research Foundation CRC1292/2 (Project No. 318346496, TP21N (MPR)), and Else-Kröner-Fresenius-Stiftung (EKFS project no 2023_EKTP05 to MPR). FHH was supported by grants of the German Research Council (DFG): HE6233/15-1, project number 517204983 and HE6233/4-2, project number 320028127. F.M. was supported by the Deutsche Forschungsgemeinschaft (DFG, German Research Foundation) project number 318346496 - SFB1292/2 TP19N. The funders had no role in study design, data collection and analysis, decision to publish, or preparation of the manuscript.

**Competing interests:** F.R., S.T., M.K., F.M., M.R., and N.R. declare that they have no conflicts of interest related to the studies. FHH served as an advisor for Novartis, CTI, Celgene/BMS, Janssen, Abbvie, GSK, Merck and AOP and received research funding from Novartis, Celgene/BMS and CTI.

**Abbreviations:** DPP, differentially phosphorylated phosphosites; IPP, increased phosphorylated phosphosites; ITAM, immunoreceptor tyrosine-based activation motif; LPS, lipopolysaccharides; MDSCs, myeloid-derived suppressor cells; MMP, matrix metalloproteases; PAMPs, pathogen-associated molecular patterns; PMN, polymorphonuclear neutrophils; PTK, protein tyrosine kinome; RIN, RNA integrity number; STK, serine threonine kinome; TAMs, tumor-associated macrophages; TANs, tumor-associated neutrophils; TLR, toll-like receptor; TREM-1, triggering receptor expressed on myeloid cells 1.

## Introduction

The immune system consists of a complex and dynamic network of cellular and molecular processes and plays a critical role in inflammatory processes and cancer [1]. Invasion by pathogenic microorganisms leads to activation of the innate immune response involving cellular components like neutrophils, monocytes, and macrophages, and molecular components like complement, cytokines, and acute phase proteins to protect the host from infections. The immediate innate inflammatory response is followed by the activation of the adaptive immune system including antigen specific T and B lymphocytes contributing to infection control and memory [2]. Dysregulated immune responses can lead to recurrent infections or autoimmunity and is associated with an increased risk of developing malignancies [3,4]. Polymorphonuclear neutrophils (PMN) are key players in activation and regulation of the innate and adaptive immune system and regulated by receptors recognizing various cytokines or pathogen associated molecular pattern (PAMPs) like toll-like receptors (TLR) [5].

In this context, the triggering receptor expressed on myeloid cells (TREM)-1, a member of the immunoglobulin (Ig) superfamily is an activating receptor expressed on PMN and monocytes amplifying the inflammatory response [6]. Since its identification, a growing number of regulatory mechanisms and potential ligands have been described [7–9]. There is a growing body of evidence of TREM-1 involvement in inflammatory disorders and cancer. TREM-1 has been upregulated in several infectious and noninfectious inflammatory diseases like lesions caused by bacteria or fungi, or inflammatory bowel disease [10–12]. Further, its soluble form sTREM-1 released by enzymatic surface shedding by matrix metalloproteases (MMP) has been identified as a marker of TREM-1 activation correlating with disease severity [13,14]. Recent data indicate a substantial role of TREM-1 in altering the course of malignancies. TREM-1 has been associated with an adverse outcome of lung cancer, HCC, breast tumors, colorectal cancer, and renal cell cancer, possibly caused by an alteration of the tumor associated neutrophils (TAN) and macrophages (TAM) [15–19]. Moreover, Ajith et al. identified decreased frequency and immunosuppressive capacity of myeloid-derived suppressor cells (MDSCs) after TREM-1 inhibition [20]. However, Juric et al. have found TREM-1 activation of myeloid cells promoting antitumor activity probably by generating a proinflammatory tumor microenvironment [21]. This broad spectrum of TREM-1 associated diseases supports targeting TREM-1 as a therapeutic approach of inflammatory diseases and cancer. Blocking TREM-1 using a recombinant TREM-1-Fc fusion or TREM-1 inhibitory peptides as well as inhibition of the eCIRP-TREM-1 ligation was shown to improve the outcome of murine sepsis models or cancer [10,20,22–24]. The TREM-1 decoy receptor Nangibotide is the first TREM-1 inhibitor reaching clinical studies [25,26]. However, no FDA approved TREM-1 inhibitors are currently available.

Besides receptor modulation, targeting TREM-1 signaling serves as another therapeutic approach. TREM-1 consists of a ligand binding Ig-structure, a transmembrane structure, and a short cytoplasmatic tail associated with the adapter molecule DAP12 containing immunoreceptor tyrosine-based activation motifs

(ITAM). Activation of TREM-1 leads to receptor crosslinking and phosphorylation of DAP12 resulting in activation of various kinases and downstream in Ca2+mobilization [6]. With a high number of developed and approved kinase inhibitors being available, kinases involved in TREM-1 signaling are promising targets [27]. Studying various ITAM-containing proteins in general and signaling pathways after TREM-1 ligation, several kinases probably being involved in TREM-1 signaling have been identified by now. Initially, Src family kinases phosphorylate tyrosines within the ITAMs of DAP12 resulting in activation of spleen tyrosine kinase (Syk) and ZAP70 tyrosine kinases [28]. Downstream pathways being activated after TREM-1 ligation include Janus kinase (Jak)2, STAT, PLCγ, p38MAPK, PI3K/PKB/AKT, ERK1/2, NF-κB, and NTAL leading to production of proinflammatory cytokines [6,7,29,30]. Besides TREM-1 mediated inflammatory response, the TREM-1 signaling cascade can interact with the TLR4 pathway via Myd88 resulting in amplification of proinflammatory cytokine production [6,9]

The TREM-1 mediated signaling pathways have been investigated for many years using various methods and cell lines. However, distinct signaling cascades of TREM-1 in various cell lines have been described [7]. Neutrophils have been reported as key players in the regulation of the immune response elucidating the relevance of their systematic signaling pathway analysis after TREM-1 ligation [5]. With the aim of revealing advanced information about the TREM-1 mediated signaling cascade in human neutrophils, we investigated the protein tyrosine kinome (PTK) and the serine threonine kinome (STK) of purified human neutrophils from healthy donors after TREM-1 ligation by kinome array. We subsequently performed RNAseq to gain further insights into the signaling cascade setting the basis for the rationale design of specific inhibitors of the TREM-1 signaling network to modulate inflammatory diseases and cancer.

## Methods

Donors. Experiments involving human subjects were carried out in accordance with the Declaration of Helsinki. Human studies with healthy volunteer blood donors were approved by the Landesaerztekammer Rhineland-Palatine Ethics Committee (approval no. 2020–15379) according to institutional guidelines and were conducted with the understanding after obtaining written informed consent of all subjects. Donors were recruited between 01/04/2022 and 31/12/2023. *Donors included both male and female adults (age range 25–62 years). All donors were healthy at the time of blood donation and had no known acute or chronic inflammatory or malignant disease.*

PMN purification. PMN were isolated from fresh heparinized venous blood of healthy volunteers by density gradient sedimentation using a dextran solution (Carl Roth, Karlsruhe, Germany) to separate leukocytes followed by density gradient centrifugation (600 x g for 30 min) using Histopaque 1077 (Sigma-Aldrich, St. Louis, MO, USA) to isolate PMN as previously described [31]. The pelleted cells were treated with ammonium-chloride-potassium buffer (Thermo Fisher Scientific, Waltham, MA, USA) for lysis of residual erythrocytes and subsequently resuspended in TM3 (Iscove's medium supplemented with 3% fetal calf serum, both Sigma Aldrich) followed by checking for purity (CD11b and CD66b coexpression, average > 95%) by flow cytometry using fluorescent monoclonal antibodies (PB anti-human CD11b, clone M1/70, BioLegend, San Diego, CA, USA and FITC anti-human CD66b, clone 80H3, Beckman Coulter, Krefeld, Germany). The purification processes and storage took place at room temperature to avoid PMN activation.

PMN activation and quality check. After each purification, PMN were investigated for activation and degranulation by flow cytometry to exclude preactivated PMN as previously described [31]. Therefore, PMN were treated with LPS (lipopolysaccharides from Salmonella enterica serotype Typhimurium (Sigma-Aldrich) at a final concentration of 1 µg/mL) or anti-TREM-1 compared to isotype control as well as unstimulated PMN for 30 min at 37°C. For TREM-1 ligation, 48-well or 96-well plates were coated with a mouse monoclonal anti-human TREM-1 antibody (clone 1C5) or mouse IgG1 (clone 4C9) serving as isotype matched control, both purified from hybridoma supernatants and used at 10 µg/mL PBS [32]. After activation, PMN were stained with PB-labeled anti-CD11b, FITC-labeled anti-CD66b, and APC-labeled anti-human CD62L (clone DREG-56, BioLegend). Activation and degranulation were determined as upregulation of CD11b or CD66b or as downregulation of CD62L. PMN activation or degranulation after stimulation with LPS or anti-TREM-1 (1C5) served as

inclusion criteria, and PMN activation in the medium control group (preactivation) or after treatment with the isotype (4C9) were defined as exclusion criteria [33]. After passing the quality check, PMN were used for further experiments.

Protein isolation. 750 µl TM3 containing 1.5x106 PMN were seeded into a coated 48-well plate and stimulated for 20 minutes at 37°C. Subsequently, cells were pelleted by centrifugation (600 x g for 3 minutes) and the supernatant was carefully discarded. MPERTM Mammalian Extraction Buffer (Thermo Fisher Scientific) containing HaltTM Phosphatase Inhibitor Cocktail and HaltTM Protease Inhibitor Cocktail (both 1:100, Thermo Fisher Scientific) were immediately added followed by incubation for 30 minutes at 4°C for cell lysis. The cell suspensions were pipetted up and down several times to improve the protein extraction. After completion, the lysates were divided into five aliquots per sample and immediately stored at −80°C. The work was done in a cold room when possible, to avoid kinase degradation. For protein quantification, the PierceTM 660-nm Protein-Assay-Kit (Thermo Fisher Scientific) was used according to the manufacture's recommendations.

Kinome array. Kinase activity profiling is established to predict kinases from cell or tissue lysates [34–36]. Protein samples of 3 biological replicates per condition were used for kinome activity determination by kinome arrays. Protein Tyrosine Kinase PamChip®s (PamGene, The Netherlands) were loaded with each 4 µg protein and Serine threonine Kinase PamChip®s (PamGene) were loaded with each 1 µg protein. One PamChip contains 144 (STK) or 194 (PTK) peptide sequences with at least one phosphosite that can be phosphorylated by determined kinases. Phosphorylation was quantified using the PamStation®12 (PamGene) and the PTK and STK reagent kits (PamGene), containing the required buffers and FITC-conjugated antibodies, according to the manufacturer's recommendations. The final data analysis was performed using the BioNavigator software (PamGene). Subsequently, differential phosphorylation was determined as $p < 0.05$ using a two-sided unpaired Student's t-test. The final data analysis including the phosphosite analysis was a service of PamGene.

The upstream kinase analysis (UKA) tool from the BioNavigator software was used to predict differential kinase activity based on publicly available databases specifying kinase-to-substrate relationships [37]. The UKA was a service of PamGene. Predicted kinases were defined as a median finale score (combining the significance of the results with the specificity) >1.2 according to the manufacturer's recommendations. The kinase statistics indicates the extent of activity alteration. For visualization the predicted kinase activity, the web application coral was used to indicate the median final scole and the kinase statistics (http://phanstiel-lab.med.unc.edu/CORAL/) [38].

Generation of transcriptome profiles via RNA-seq. 1000 µl TM3 containing 3x106 PMN were seeded into a coated 24-well plate and stimulated for 60 minutes at 37°C. Subsequently, cells were pelleted by centrifugation (600 x g for 3 minutes) and the supernatant was carefully discarded. RNA of 3 biological replicates per condition were used. Total RNA was isolated using Trizol (ThermoFisher Scientific). Quantity of total RNA was assessed with Qubit 2.0 and quality was checked using a RNA 6000 Nano chip on Agilent's bioanalyzer. The determined RNA integrity number (RIN) of all samples was ≥ 7. 200 ng of total RNA was send to Novogene (Cambridge, UK) for library preparation and sequencing. Sequencing strategy was paired-end 150 cycles (PE150) eith 30mio PE reads per sample. Sequence reads were processed using Qiagen's software CLC Genomics workbench (v22.0.1) with CLC's default settings for RNA-Seq analysis (Reference sequence = Homo sapiens (GRCh38); Gene track = Homo sapiens (GRCh38.104); mRNA track = Homo sapiens (GRCh38.104); Use spike-in controls = no; Mismatch cost = 2; Insertion cost = 3; Deletion cost = 3; Length fraction = 0,8; Similarity fraction = 0,8; Global alignment = No; Strand specific = Both; Library type = Bulk; Maximum number of hits for a read = 10; Count paired reads as two = No; Ignore broken pairs = Yes; Expression value = TPM). Reads were aligned to GRCh38 genome. Expression value unit is TPM. Differential expression analyses was performed using the Empirical analysis of differential gene expression (EDGE) tool, implementing the `Exact Test' for two-group comparisons developed by Robinson and Smyth for situations in which only a few replicates are available, in the CLC-GWN estimating tag-wise dispersions [39]. Differentially expressed genes were defined as false discovery rate (FDR) adjusted p value <0.05, fold change (FC) <−2 or >2, and experiment difference of gene expression < −4 or >4.

Bioinformatics. For pathways and ontologies analysis, the gene set library WikiPathway 2021 Human was analyzed using the publicly available web-based tool Enrichr (https://maayanlab.cloud/Enrichr/) [40–42]. Therefore, differentially

                                                                 

expressed genes or phosphorylated phosphosites were separated in up and downregulated and subsequently used for enrichment analysis. Before adding to Enrichr input list, UniPROT IDs were converted to Gene names using the dedicated tool on the uniprot website (https://www.uniprot.org/id-mapping/) [43].

## Results

### Characterization of the kinome activity in human PMN upon TREM-1 or TLR4 ligation

TREM-1 signaling via Src and Syk family kinases resulting in activation of JAK2, p38MAPK, PI3K/PKB/AKT, and ERK1/2 has been previously described. With the aim of gaining deeper insights into the TREM-1 mediated kinome activity in human neutrophils, we performed a global kinase activity profiling. As TREM-1 interacts with TLR signaling, we additionally used LPS being a TLR4 agonist for subsequent identification of TREM-1 specific kinome activity. In a first step, we generated protein lysates of purified human PMN 20 minutes after TREM-1 or TLR4 ligation. We subsequently determined their phosphorylation capacities of PTK and STK arrays using a PamStation 12. After data analysis, we identified the differentially phosphorylated phosphosites (DPP) by comparing TREM-1 vs. isotype matched control to identify the TREM-1 induced alteration of the global kinome acvitity. Out of the PTK, we revealed 63 (Fig 1 (a)) and out of the STK, we revealed 29 (Fig 1 (b)) differentially phosphorylated sites. With 63/194 (about 32%), we identified more TREM-1 specific phosphorylation alteration of the PTK compared to the STK with 29/144 (about 20%). In both PTK and STK, we did not observe any TREM-1 specific phosphorylation downregulation. Besides TREM-1 induced kinome activity, we investigated the TLR4 induced alteration of the kinome activity by comparing LPS treated PMN vs. medium control resulting in 58 differentially phosphorylated PTK phosphosites (Fig 1 (c)) and 42 differentially phosphorylated STK phosphosites (Fig 1 (d)). Within STKs, we identified 5 phosphosites with decreased phosphorylation. In the next step, we compared the DPP after TREM-1 ligation with those after TLR4 ligation to identify mapping as well as differentiating phosphorylation tendencies. Out of the increased phosphorylated phosphosites, we identified 58 (about 45%) mapping phosphosites and 34 (about 26%) TREM-1 specific as well as 37 (about 29%) LPS specific increased phosphorylated phosphosites (Fig 1 (e)). TLR4 ligation led to decreased phosphorylation of 5 phosphosites whereas TREM-1 ligation exclusively showed increased phosphorylation resulting in non-existence of mapping terms out of the decreased phosphorylated phosphosites (Fig 1 (f)). Overall, we mainly found increased kinome activity after TREM-1 as well as after TLR4 ligation confirming TREM-1 as an activating receptor. Partially, TREM-1 and TLR4 ligation resulted in mapping kinome activity elucidating the cross-linking of the TREM-1 and TLR4 signaling cascade.

PMN were seeded into a cell culture plate in presence or absence of LPS or coated with anti-TREM-1 (1C5) or an isotype matched control mAb (4C9), respectively, for 20 minutes for subsequent kinome array. Therefore, lysates of n = 3 biological replicates from various healthy donors and individual experiments were used. The phosphorylation of various phosphosites was quantified and analyzed using a PamStation 12 and the BioNavigator software. Differential phosphorylation was determined by comparing anti-TREM-1 (1C5) vs. isotype control (4C9) or LPS vs. medium control and significance was defined as p < 0.05 using a two-sided unpaired Student's t-test. The differential phosphorylation of the PTK phosphosites (a) and the STK phosphosites (b) after TREM-1 ligation as well as after TLR4 ligation (PTK: c and STK: d) are indicated as volcano plots. Significant results are colored red. From the significantly differentially phosphorylated phosphosites (DPP), increased phosphorylated phosphosites after TREM-1 ligation were compared with increased phosphorylated phosphosites after TLR4 ligation (e) and decreased phosphorylated phosphosites after TREM-1 ligation were compared with decreased phosphorylated phosphosites after TLR4 ligation (f) and mapping phosphorylation tendencies are indicated as Venn diagrams.

### Kinome pathway analysis upon TREM-1 or TLR4 ligation

After identification of the TREM-1 mediated kinome activity, we performed pathway analysis from the DPP using the WikiPathway database for further biological interpretation. As we could not find relevant decreased phosphorylation

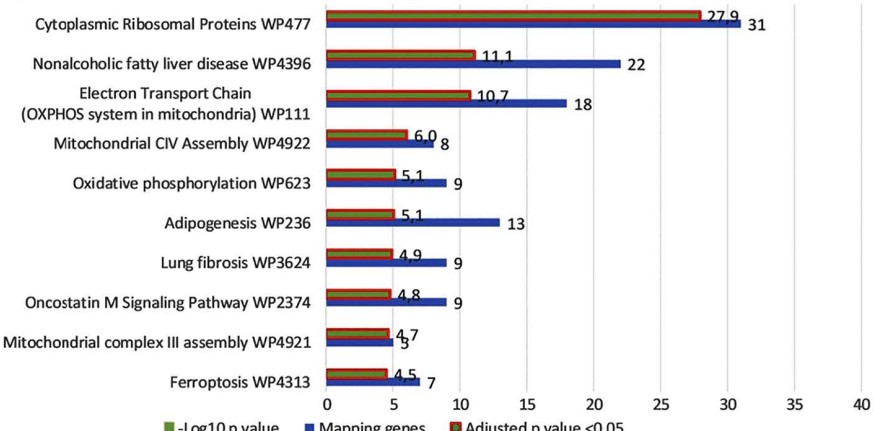

**a) Upregulated pathways after TREM-1 ligation**

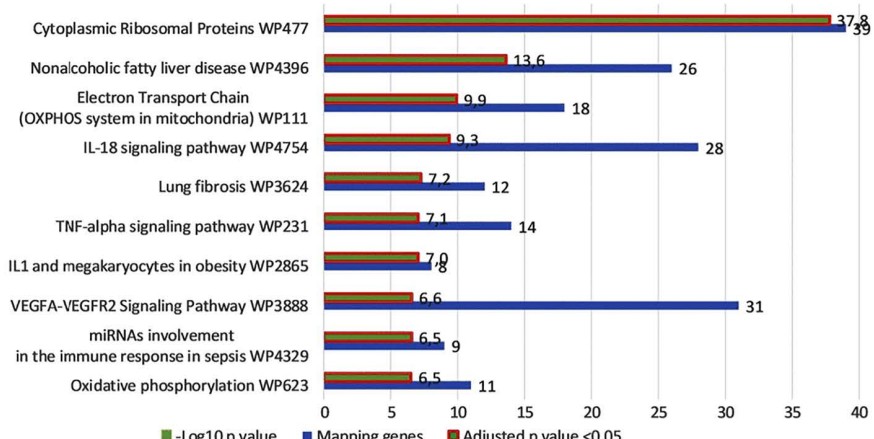

**b) Upregulated pathways after TLR4 ligation**

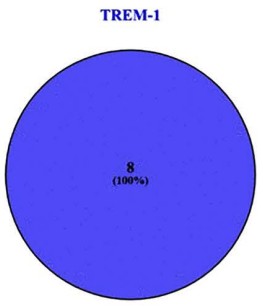

**c) Upreguated pathways**

**d) Downregulated pathways**

**Fig 1. Differential phosphorylation of phosphosites after TREM-1 or TLR4 ligation.**

activity, we restricted the further analysis to the increased phosphorylated phosphosites (IPP). Therefore, the significant results of the PTK and STK (Fig 1 (a-d): red spots) were used for Enrichr analysis after converting UniPROT IDs into gene names. We found 146 significantly enriched terms after TREM-1 ligation and 211 after TLR4 ligation containing several intracellular protein and kinase signaling pathways. Out of them, the top ten enriched pathways after TREM-1 ligation with Ras signaling and PI3K-Akt signaling pathway being the top enriched terms (Fig 2 (a)) and after TLR4 ligation, respectively (Fig 2 (b)) are indicated. Both Ras-ERK-MAPK and PI3K-Akt signaling pathways have been found to be involved in TREM-1 signaling validating our results [44]. Subsequently, we checked for mapping and differentiating terms by comparing the two treatment conditions. The comparison resulted in 138 mapping terms (about 63%), 8 (about 4%) TREM-1 specific, and 73 (about 33%) TLR4 specific terms (Fig 2 (c)). Out of the TREM-1 specific enriched terms, PPAR-α pathway (WP2878), prostaglandin synthesis and regulation (WP98), adipogenesis (WP236), effects of nitric oxide (WP1995), and insulin signaling (WP3635 and WP3634) represent selected pathways. A table containing the enriched pathways from the

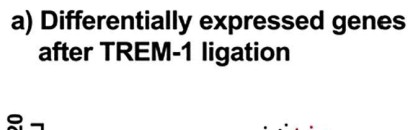
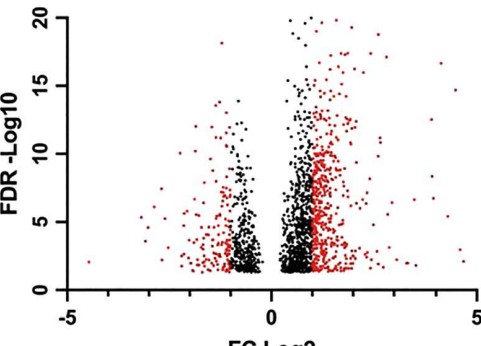
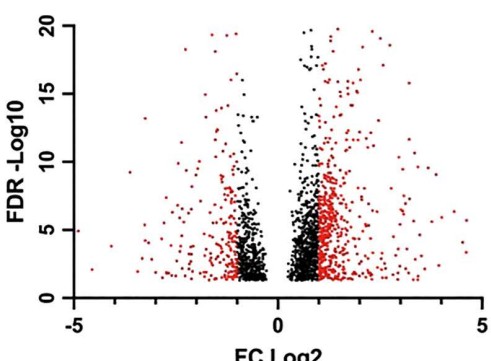
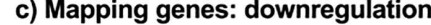
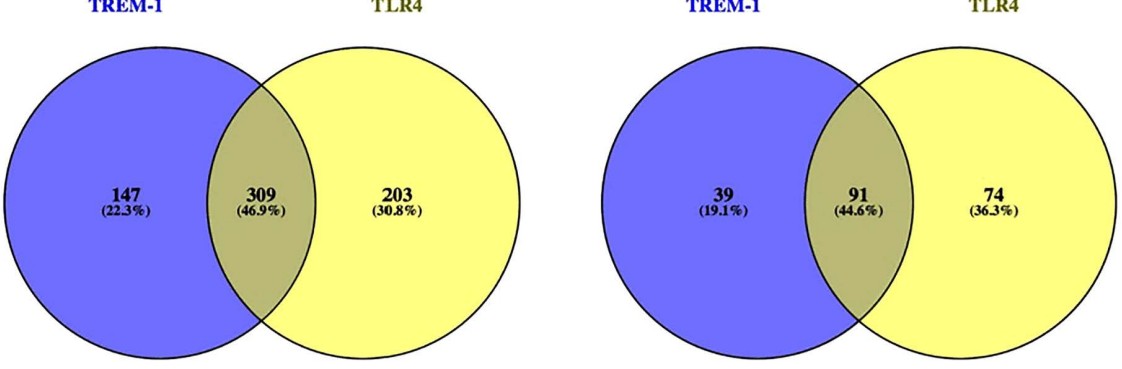

**Fig 2. Pathway analysis pf increased phosphorylated phosphosites.**

IPP is supplemented (SI1). Taken together, we revealed signaling pathways being active after TREM-1 as well as TLR4 ligation and identified TREM-1 specific pathways linking TREM-1 to various biological processes or diseases.

Kinome profile (consisting of n = 3 biological replicates from various healthy donors and individual experiments; Fig 1) was analyzed using Enrichr. Prior to Enrichr analysis, UniPROT IDs were converted to gene names using uniprot. Enriched terms after TREM-1 ligation (a) and after TLR4 ligation (b) were identified from the WikiPathway 2021 Human database out of the Pathways category. The top 10 enriched terms are displayed based on the –log10 p value. Significance was defined as adjusted p value <0.05 and is indicated as * next to the p value. Enriched terms after TREM-1 ligation were compared with enriched terms after LPS treatment (c) and mapping pathways are indicated as Venn diagrams.

## Kinase prediction after TREM-1 or TLR4 ligation

Having analyzed the TREM-1 mediated kinome activity as well as several pathways and ontologies from the DPP, we wanted to get further insights into the involved kinases. Based on the DPP, upstream kinase analysis (UKA) was performed to predict activated kinases. We predicted 31 PTK (Fig 3 (a)) and 38 STK (Fig 3 (b)) being after TREM-1 ligation. The kinase statistic ranged from 2.4 to 5.5 (PTK) and from 1.2 to 1.6 (STK) suggesting TREM-1 mediated kinase alteration of the PTK to a greater extent than the STK. Out of the PTK, ten predicted kinases belong to the Src family kinases with Lck, Src, and Lyn being the top 3 predicted kinases. After the Src family kinases, the Syk family kinases Syk and ZAP70 followed by the the Abl family kinases Arg and Abl as well as the InsR family kinases InSR and IGF1R and the PDGFR family kinases FLT3, PDGFRα, and Kit belong to the predicted PTK (Fig 3 (a)). The predicted STK can be categorized into kinase groups obtaining a more comprehensible description. The most frequently predicted STK belong to the AGC group containing mainly PKKC, RSK, PKA, and PKG family kinases followed by the CAMK group containing mainly PIM and MAPKAPK family kinases as well as the CMGC group containing mainly CDK family kinases. In more detail, two kinases belong to the PKG family kinases with PKG2 and PKG1 being placed in the top 3 predicted kinases. Beside the PKG family kinases, we identified 15 more kinases out of the AGC group with PKAα, PKCδ, p70S6Kδ, PKCα, and PRKX being placed in the top 15 predicted kinases. In addition to the AGC group kinases, we identified 10 CAMK group kinases with AMPKα1, Pim1, Pim2, and Pim3 being placed in the top 15 predicted kinases as well as 5 CMGC group kinases (Fig 3 (b)). The UKA from the DPP after TLR4 ligation resulted in 35 predicted PTK (Fig 3 (c)) and 61 predicted STK (Fig 3 (d)). Subsequently, we compared the predicted kinases after TREM-1 ligation with those after LPS treatment resulting in 49 (about 42%) mapping kinases. Moreover, we identified 20 (about 17%) TREM-1 specific and 47 (about 41%) TLR4 specific kinases (Fig 3 (e)). The top five TREM-1 specific predicted PTK consist of FRK, FLT3, Srm, Fes, and BTK, and the top five TREM-1 specific predicted STK consist of PKG2, PKAα, PKG1, AMPKα1, and PKCδ (Fig 3 (f)). A table containing all predicted PTK and STK is supplemented (SI2). In sum, we predicted individual kinases being active after TREM-1 or TLR4 ligation and identified TREM-1 specific kinases serving as potential targets to modulate TREM-1 signaling.

The UKA tool from the BioNavigator software was used to predict differential kinase activity from the DPP. Therefore, the kinases after UKA were ranked by Median Final Score which is based on the specificity and the significance of the results and correlates with the probability of being differently activated. A median final score >1.2 was defined as predicted kinase. Out of the predicted kinases, the top 15 kinases are named. The predicted PTK (a) and STK (b) after TREM-1 ligation and the predicted PTK (c) and STK (d) after TLR4 ligation, respectively are displayed. The predicted kinases are displayed based on their Mean Specificity Score (x-axis) indicating the specificity of the alteration of kinase activity, Median Kinase Statistic (y-axis) indicating the extent of activity alteration (<0 means inhibition, >0 means activation), Median Final score (bubble size), and their related kinase family (bubble color). Predicted kinases after TREM-1 ligation were compared with predicted kinases after TLR4 ligation and mapping kinases are indicated as Venn diagram (e). Out of the TREM-1 specific predicted kinases (e: blue circle), the top 5 predicted PTK and STK are displayed based on their Median Final Score (f).

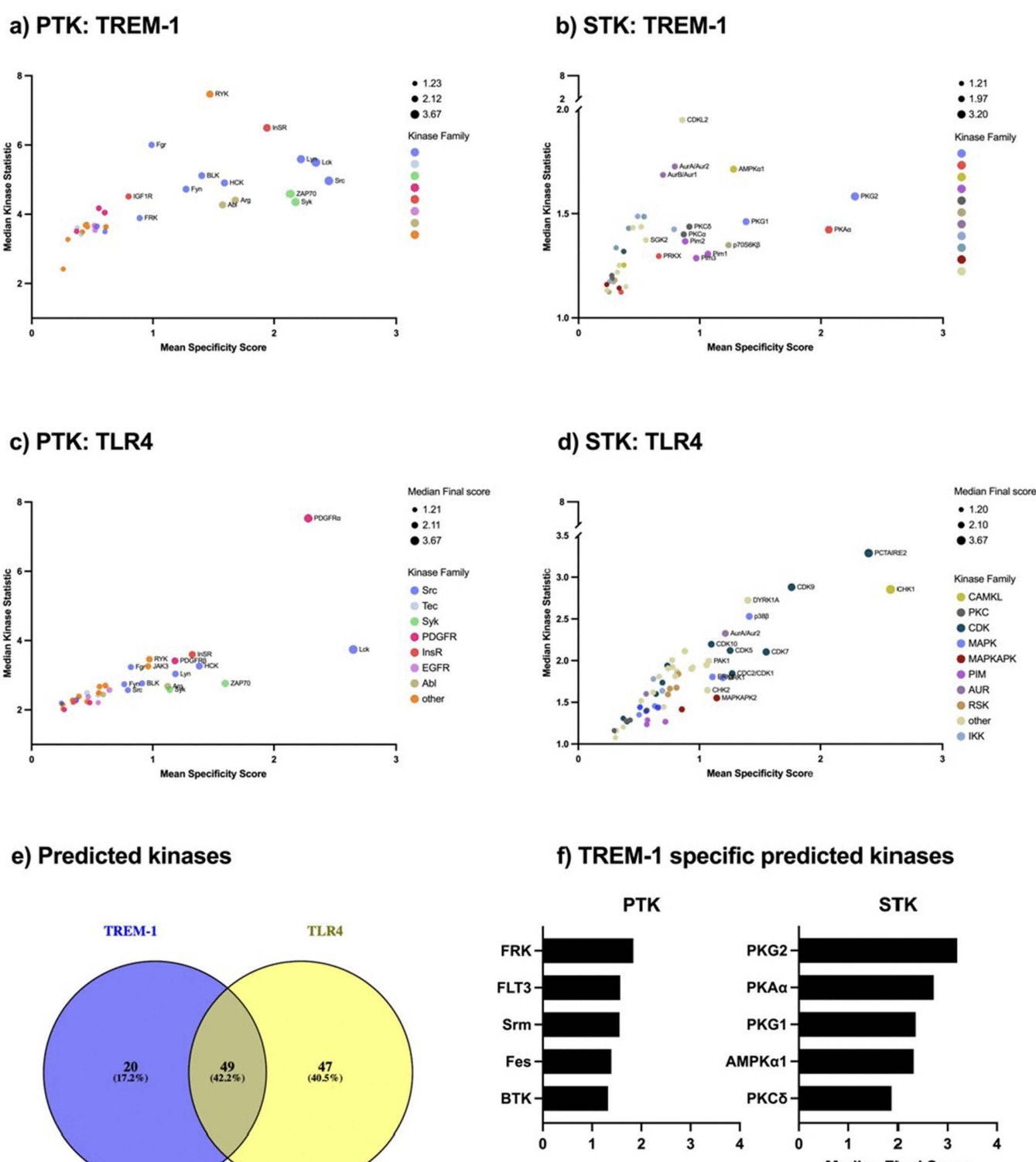

**Fig 3. Upstream Kinase Analysis (UKA) from the differentially phosphorylated phosphosites.**

## Characterization of the transcriptomic profile of human PMN upon TREM-1 or TLR4 ligation

After uncovering the global TREM-1 mediated kinome activity as well as predicting individual kinase activities, we performed RNAseq to gain further insights into the TREM-1 mediated signaling cascade. As cellular effects on transcriptomic level are delayed in time, we isolated RNA 60 minutes after TREM-1 ligation. After sequencing and data analysis, we identified 586 differentially expressed genes after TREM-1 ligation (Fig 4 (a), red spots) and 677 differentially expressed genes after TLR4 ligation (Fig 4 (b), red spots). TREM-1 ligation led to upregulation of 456 genes and downregulation of 130 genes indicating TREM-1 mediated increasing effects on gene expression levels. With upregulation of 512 and downregulation of 165 genes, TLR4 ligation also increased genes expression. We then compared the differentially expressed genes after TREM-1 ligation with those after TLR4 ligation to identify mapping and differentiating genes. Out of the upregulated genes, we found 309 (about 47%) mapping genes, 147 (about 22%) TREM-1 specific, and 203 (about 31%) TLR4 specific genes. Out of the downregulated genes, we found 91 (about 47%) mapping genes, 39 (about 19%) TREM-1 specific, and 74 (about 36%) TLR4 specific genes. In contrast to our kinome data, we found upregulated but also downregulated gene expressions after TREM-1 or TLR4 ligation elucidating their ability of controlling complex cell processes. However, we found distinct similarities between the TREM-1 and TLR4 mediated transcriptome that, in line with our kinome data, clarifies the cross-linking of the TREM-1 and TLR4 signaling cascade.

PMN were seeded into a cell culture plate in presence or absence of LPS or coated with anti-TREM-1 (1C5) or isotype matched control (4C9), respectively, for 60 minutes for subsequent RNAseq. Lysates of n = 3 biological replicates from one healthy donor of 3 individual experiments were used. The gene expression levels of anti-TREM-1 (1C5) vs. isotype control (4C9) or LPS vs. medium were compared and significance was defined as FDR adjusted p value <0.05. The results are displayed based on their FDR and tagwise dispersions FC. Out of them, differently expressed genes were defined as FC > 2 and are colored red. Differentially expressed genes after TREM-1 ligation (a) and after TLR4 ligation (b) are indicated as volcano plots. Upregulated genes after TREM-1 ligation were compared with upregulated genes after TLR4 ligation (c), downregulated genes after TREM-1 ligation were compared with downregulated genes after TLR4 ligation (d), and expression overlaps are indicated as Venn diagrams.

Proceeding as before with the analysis of the DPP, we performed pathway analysis from the differentially expressed genes also using the WikiPathway 2021 Human database. Differing from the kinomics, we found upregulated but also downregulated genes leading to two enriched terms clusters. Our pathway analysis revealed 19 significantly enriched terms from upregulated genes after TREM-1 ligation. Out of the top 10 enriched pathways, the mapping genes ranged from 31 to 5 and the –Log10 p value from 27.9 to 4.5 (Fig 5 (a)). The pathway analysis from downregulated genes after TREM-1 ligation resulted in 8 significantly enriched terms. However, the mapping genes ranged from 5 to 2 and the –Log10 p value from 4.1 to 2.7 being distinctly less compared to the enriched pathways from upregulated genes indicating TREM-1 being an activating receptor (SI3). TLR4 ligation led to 58 significantly enriched pathways from upregulated genes with mapping genes ranging from 39 to 8 and –Log10 p value from 37.8 to 6.5 out of the top 10 enriched pathways (Fig 5 (b)). In contrast to TREM-1, the pathway analysis from downregulated genes did not show any significantly enriched pathway (SI3). Fig 5a shows the top 10 enriched terms after pathway analysis of upregulated genes after TREM-1 ligation containing several terms associated with protein translation (WP477) or generation and regulation of bioenergetics (WP111, WP4922, WP623, and WP4921). Moreover, we found disease-associated terms dealing with fatty liver disease (WP4396), adipogenesis (WP236), and lung fibrosis (WP3624) as well as pathways associated with oncostatin m signaling (WP2374) and ferroptosis (WP4313). The pathway analysis from the downregulated genes after TREM-1 ligation revealed reduced transcription regulation as well as downregulation of several immune receptor signaling pathways and regulation (WP3877, WP455, WP3858, WP4868, and WP3945) and G protein-coupled receptors (WP455 and WP80; SI3). Fig 5b shows the top 10 enriched pathways from upregulated genes after TLR4 ligation. Like the pathways after TREM-1 ligation, we found terms associated with protein translation (WP477) and bioenergetics (WP111 and WP623) as well as disease-associated terms (WP4396 and WP3624). Moreover, we found terms associated with several immune

## a) Enriched pathways after TREM-1 ligation

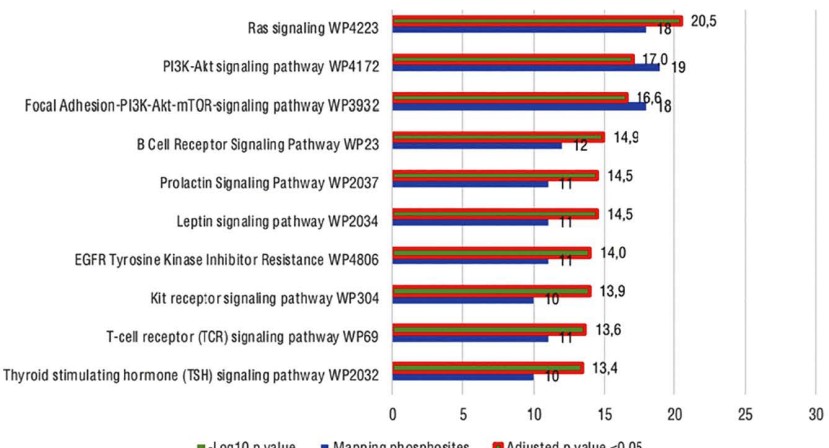

- -Log10 p value
- Mapping phosphosites
- Adjusted p value <0.05

## b) Enriched pathways after TLR4 ligation

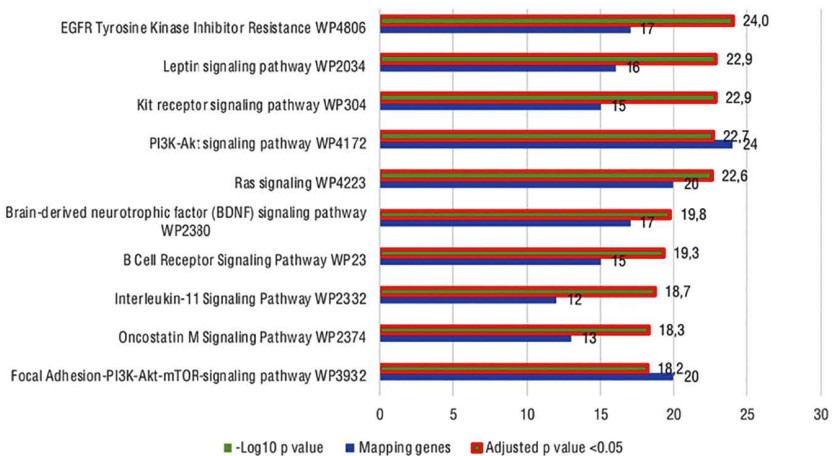

- -Log10 p value
- Mapping genes
- Adjusted p value <0.05

## c) Mapping pathways

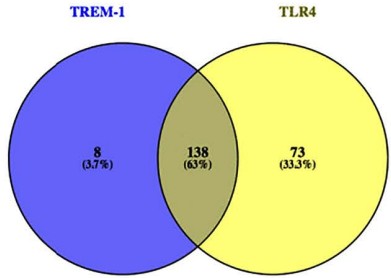

TREM-1    TLR4

8 (3.7%)    138 (63%)    73 (33.3%)

**Fig 4. Differently expressed genes after TREM-1 or TLR4 ligation.**

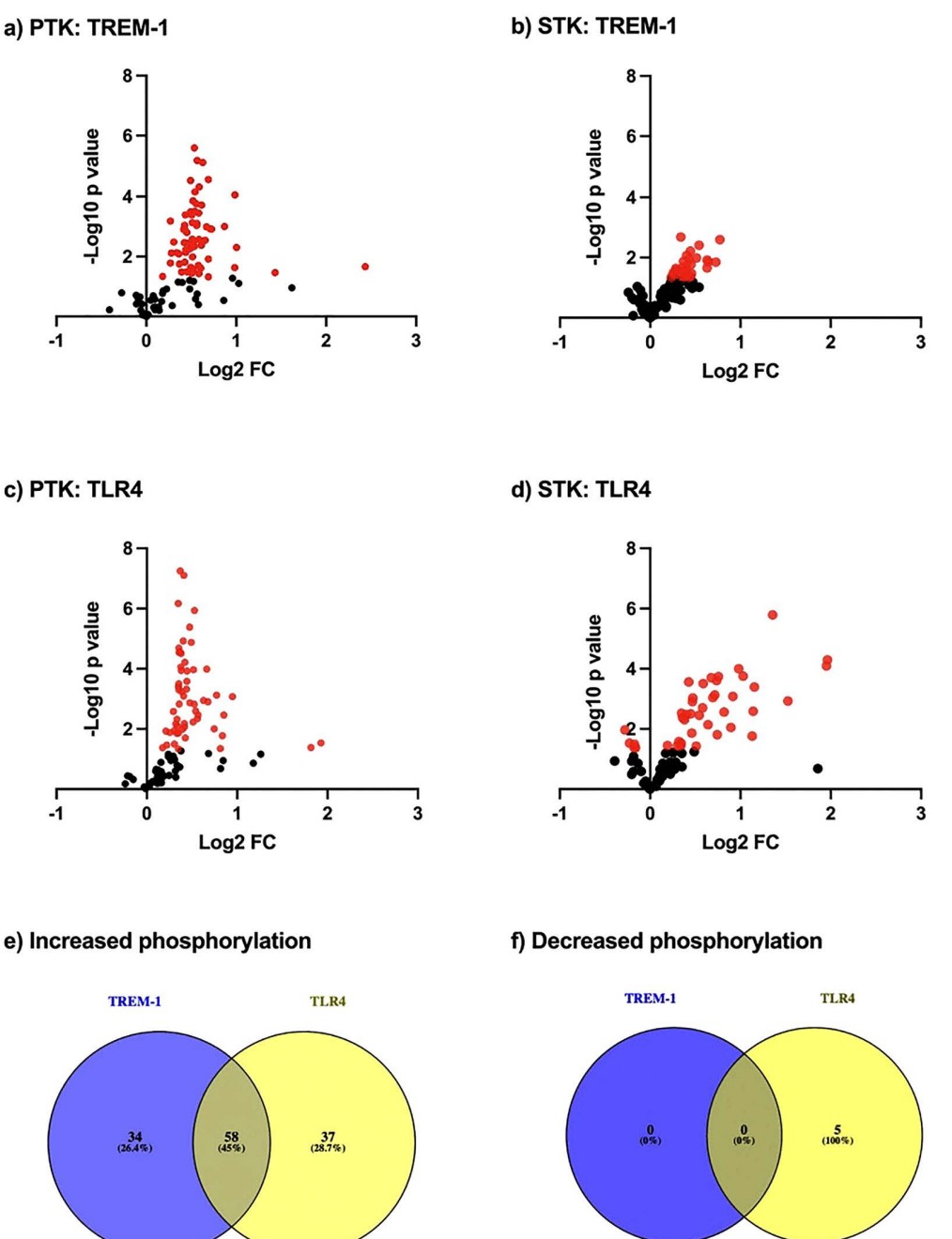

**Fig 5. Pathway analysis of upregulated genes after TREM-1 and TLR4 ligation.**

system signaling pathways and sepsis (WP4754, WP231, WP2865, WP3888, and WP4329). A table containing the enriched pathways from the differentially expressed genes is supplemented (SI3). Finally, we identified mapping and differentiating terms from upregulated and downregulated genes, respectively. From the upregulated pathways, the comparison resulted in 16 (about 26%) mapping terms, 3 (about 5%) TREM-1 specific, and 42 (about 69%) LPS specific terms

(Fig 5 (c)). Going into detail, the TREM-1 specific pathways are associated with adipogenesis (WP4211) and amyotrophic lateral sclerosis (ALS, WP2447). As LPS treatment did not cause gene downregulation leading to significantly enriched pathways, we only found 8 TREM-1 specific terms from downregulated genes (Fig 5 (d)). Taken together, we identified significantly enriched pathways clarifying the involvement of TREM-1 and TLR4 in various biological processes and several diseases.

Gene profile 60 minutes after TREM-1 ligation or LPS treatment was analyzed for enriched pathways using Enrichr and the WikiPathway 2021 Human database. The top 10 enriched terms from TREM-1 ligation (a) and TLR4 ligation (b) are displayed based on their –Log10 p value with additionaly indicated number of mapping genes (number of associated genes to that specific gene set). Significance was defined as adjusted p value <0.05 and is indicated as a red square around the bar. Enriched pathways from upregulated genes (c) or from downregulated genes (d), respectively, were compared and mapping pathways are indicated as Venn diagrams.

## Discussion

PMN are key players in activation and regulation of the immune response and as TANs, being part of the tumor microenvironment, they influence tumor growth and metastasis [5,45]. Being highly expressed on PMN, monocytes, and macrophages, TREM-1 plays a crucial role in acute and chronic infectious and non-infectious inflammatory disorders and receives increasing attention in several cancer conditions [46]. TREM-1 expression is associated with adverse tumor outcomes probably by alteration of the TANs and TAMs potentially serving as a novel target in cancer treatment [15–19]. By now, several aspects of the TREM-1 signaling cascade including various kinases have been described. However, the TREM-1 signaling cascade is cell type specific complicating its understanding [7,46]. In this study, we characterized the TREM-1 signaling cascade of human neutrophils by kinome array and RNA-seq. Besides validating known kinases being involved in TREM-1 signaling, we could predict activity of several kinases previously not described after TREM-1 ligation. Based on our transcriptomic data, we validated TREM-1 as an activating receptor and connected it to several diseases and biological processes.

Using kinase activity profiling, we determined the phosphorylation of 196 (PTK) and 144 (STK) various peptide sequences making us able to systematically analyze the kinome in human PMN after TREM-1 ligation [34–36]. We predicted a high number of PTK and STK after TREM-1 ligation and revealed several signaling pathways. The top predicted PTK include Src and Syk family kinases being in line with current literature [46]. Moreover, we confirmed the RAS/MAPK as well as the PI3K-Akt-mTOR signaling pathway with our top enriched pathways after TREM-1 ligation [44,46].

We additionally analyzed the kinome and gene expression profile after TLR4 ligation by LPS treatment. As a TLR4 agonist being involved in infectious inflammatory diseases and sepsis mainly caused by Gram-negative bacteria, LPS served as a positive control for PMN activation [47,48]. Functional TLR4 is essential for wound healing, controlling bacterial infections, and is involved in anti-tumor T cell immune response [49–51]. Hyperactivation of TLR4 is associated with several diseases like sepsis or acute lung injury explaining the approaches of TLR4 targeting [52]. As TREM-1 interacts with TLR4, the identification of TREM-1 specific signaling pathways facilitates the determination of potential TREM-1 signaling targets [53,54]. The comparison of the results after TREM-1 ligation with LPS treatment made us available to define TREM-1 specific kinase activities as well as TREM-1 specific gene expressions.

Our kinome analysis revealed TREM-1 specific dependency of pathways associated with adipogenesis and lipid metabolism which could be validated by our transcriptomic data. In line with that, we found genes associated with nonalcoholic fatty liver disease. TREM-1 involvement in fatty liver disease and liver fibrosis has been described previously. TREM-1 was shown to mediate liver inflammation through NF-κB and PI3K/AKT pathways, also being validated by our kinome analysis, resulting in fatty liver disease [55]. Nguyen-Lefebvre et al. attributed TREM-1 dependent liver inflammation and fibrosis to monocytes and monocyte-derived macrophages and Kupffer cells [56]. However, current literature describes neutrophils contribution in genesis of fatty liver disease [57]. Tornai et al. demonstrated improved course of alcoholic fatty

liver disease through TREM-1 inhibition going along with reduced macrophage and neutrophil recruitment and activation [58]. Our findings complement to the previous description suggesting a PMN mediated contribution to fatty liver disease through TREM-1 activation.

Apart from the liver, fibrosis can affect any organ and is often associated with chronic inflammatory disorder mediated by local tissue fibroblasts but also by inflammatory monocytes, macrophages, and neutrophils [59,60]. Moreover, alternatively activated macrophages (M2) are associated with increased fibrosis and serve as potential targets in treating lung fibrosis [61–63]. Xiong et al. showed that TREM-1 inhibition reduced fibrosis in a mouse model of bleomycin induced pulmonary fibrosis and attributed the pro-fibrotic effect of TREM-1 to alveolar epithelial cell senescence [64]. However, the function of TREM-1 expressed on inflammatory myeloid cells in the context of lung fibrosis remains unclear. We observed TREM-1 mediated upregulation of genes associated with lung fibrosis indicating a TREM-1 mediated contribution of myeloid cells to lung fibrosis.

Besides several genes associated with cellular metabolism, we found TREM-1 specific association with ALS. Based on current literature, TREM-1 being expressed on microglia is involved in the pathogenesis of various central nervous system diseases like ischemic stroke, intracerebral hemorrhage (ICH), Parkinson's disease or Alzheimer's disease and is associated with poorer prognosis [65]. Moreover, Lu et al. demonstrated TREM-1 mediated microglia polarization via PKCδ resulting in enhanced neuroinflammation in ICH [66]. In line with that, we found TREM-1 specific PKCδ activation in our kinome studies. Microglial activation through TREM-2 is associated with ALS, however, the pathogenesis is hardly understood [67]. Actually, TREM-1 has not been linked clearly to ALS [68]. Our findings indicate a contribution of TREM-1 to the genesis of ALS.

Alteration of the TAM polarization or MDSCs are possible mechanisms of TREM-1 influencing tumor growth and outcome [15,20]. Chen et al. showed that downregulation of TREM-1 favored a M1 phenotype through inhibition of PI3K/AKT/mTOR signaling resulting in suppressed migration and invasion of liver cancer cells [16]. In line with that, we found PI3K/AKT/mTOR signaling after TREM-1 ligation. Kim et al. found that PMN-MDSC spontaneously died by ferroptosis limiting the T cell functionality [69]. Ferroptosis is a newly identified type of cell death dependent on lipid peroxidation and its induction can improve tumor cell killing in vitro [70]. However, Kim et al. showed that the immunosuppressive effect of ferroptosis in PMN-MDSCs promoted tumor growth [69]. As we found genes associated with ferroptosis in PMN after TREM-1 ligation, TREM-1 induced ferroptosis in PMN possibly contributes to adverse cancer progression.

Out of the kinases specifically activated after TREM-1 ligation, BTK is required for TREM-1 signaling [71]. Moreover, BTK involvement in LPS mediated TLR4 signaling as well as in neutrophils functionality in general has been described suspecting the TREM-1 specificity of BTK [72–74]. However, Pérez de Diego et al. found functional BTK-null monocytes and PMN in patients with X-linked agammaglobulinemia and concluded different BTK dependency between human and murine myeloid cells [75]. Furthermore, from patients receiving the BTK inhibitor ibrutinib, Stadler et al. found decreased functionality of PMN after TREM-1 ligation whereas the LPS mediated PMN effector functions remained functional [33]. In line with that, our findings indicate a more important role of BTK in TREM-1 signaling than in LPS signaling.

Altogether, we predicted several kinases being active after TREM-1 ligation on PMN. As TREM-1 signaling differs between several cell types, we could link the kinases to TREM-1 signaling in human PMN [7]. Several predicted kinases have not previously been associated with TREM-1 constituting new aspects. For instance, to our knowledge, FLT3 has not previously been linked to TREM-1. However, FLT3 is explicitly investigated in the context of FLT3 mutant AML and with midostaurin, gilteritinib, and quizartinib, several FLT3 inhibitors have been approved for FLT3 targeted AML therapy [76–78]. Currently, a high number of various FDA approved kinase inhibitors (2023: 72) are available [27]. Finally, we linked the TREM-1 mediated kinome activity to the resulting transcriptome and revealed hints of TREM-1 contribution to several diseases and biological processes. This study facilitates selecting kinase inhibitors for further validation in vitro and in vivo with the aim of targeting TREM-1 signaling in various inflammatory or cancer disease conditions.

## Author contributions

**Conceptualization:** Florian Heidel, Markus Philipp Radsak.

**Data curation:** Markus Philipp Radsak.

**Funding acquisition:** Markus Philipp Radsak.

**Investigation:** Markus Philipp Radsak.

**Methodology:** Frederic Ries, Matthias Klein, Nora Rogmann, Sophie Többen, Federico Marini, Florian Heidel.

**Resources:** Matthias Klein.

**Writing – original draft:** Frederic Ries, Markus Philipp Radsak.

**Writing – review & editing:** Markus Philipp Radsak.

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
