## [Decision Letter · Decision Letter 0]

8 Dec 2025

Dear Dr. Radsak,

Thank you for submitting your manuscript to PLOS ONE. After careful consideration, we feel that it has merit but does not fully meet PLOS ONE’s publication criteria as it currently stands. Therefore, we invite you to submit a revised version of the manuscript that addresses the points raised during the review process.

We look forward to receiving your revised manuscript.

Kind regards,

Qi Wu

Academic Editor

PLOS One

Journal Requirements:

“This work was supported by the Deutsche Forschungsgemeinschaft, (German Research Foundation): grant CRC1066 TP B18 (MPR), German Research Foundation CRC1292/2 (Project No. 318346496, TP21N (MPR)), and Else-Kröner-Fresenius-Stiftung (EKFS project no 2023_EKTP05 to MPR). FHH was supported by grants of the German Research Council (DFG): HE6233/15-1, project number 517204983 and HE6233/4-2, project number 320028127. F.M. was supported by the Deutsche Forschungsgemeinschaft (DFG, German Research Foundation) project number 318346496 - SFB1292/2 TP19N.”

Additional Editor Comments:

In particular, both reviewers highlighted the lack of functional assessment/validation.

Reviewer's Responses to Questions

**Comments to the Author**

1. Is the manuscript technically sound, and do the data support the conclusions?

Reviewer #1: Yes

Reviewer #2: Yes

2. Has the statistical analysis been performed appropriately and rigorously?

Reviewer #1: No

Reviewer #2: Yes

3. Have the authors made all data underlying the findings in their manuscript fully available?

Reviewer #1: Yes

Reviewer #2: Yes

4. Is the manuscript presented in an intelligible fashion and written in standard English?

Reviewer #1: Yes

Reviewer #2: Yes

Reviewer #1: The authors have compiled an interesting manuscript characterizing TREM1 signaling in healthy human neutrophils. The paper is generally well written, and the data presented are relevant. However, the study still lacks several crucial data points that should be addressed to strengthen clarity and reproducibility.

1. Donor characteristics require clarification. In the Methods section, please expand the details regarding the human donors, including age, sex, and any other relevant information. This is important, as TREM1 has been associated with age-related immune responses in myeloid cells, including neutrophils. Additionally, the use of n = 3 donors across all experiments is relatively weak given the demographic variability known to influence neutrophil signaling responses.

2. Typographical inconsistencies in reported cell numbers. There are multiple typographical errors throughout the manuscript, particularly in the reported cell numbers. These should be reviewed and corrected to ensure accuracy and consistency.

3. Improved annotation needed in Figure 1(a–d). In Figure 1(a–d), it would be helpful if a few of the relevant kinases were explicitly labeled or called out within the figure. This would improve readability and highlight key signaling components.

4. Difficulty separating TREM1- versus TLR4-specific signaling. In all experiments, anti-TREM1 was used for TREM1 ligation and LPS for TLR4 ligation. However, LPS can also induce TREM1 dimerization and signaling. Because of this overlap, it becomes difficult to clearly separate the influence of TLR4 from TREM1 signaling. This may also explain why, in Figure 2C, 63% of the mapped pathways are common between TLR4 and TREM1.

5. Font size adjustment needed in Figure 3(a–d). In Figure 3(a–d), please increase the font size, as the current labels are difficult to read.

6. Major conceptual challenge in distinguishing TLR4- versus TREM1-mediated pathways. A critical difficulty in this paper is separating the effects mediated by TLR4 alone versus TREM1 alone in PMNs. As shown in Figure 5A and 5B, the top three pathways upregulated after TREM1 ligation are identical to those upregulated during TLR4 ligation. The only way to truly dissect these pathways would be to knock down TLR4 or TREM1 in PMNs and compare the signaling outcomes during activation.

7. Lack of functional assessment of PMN responses. While the manuscript effectively analyzes the kinome and transcriptome of healthy PMNs, it remains unclear whether these signaling and transcriptional changes translate into functional differences in neutrophil behavior. Clarifying whether TREM1 versus TLR4 ligation leads to measurable functional alterations (e.g., cytokine release, ROS production, degranulation, migration) would strengthen the biological relevance of the findings.

Reviewer #2: The authors have done a good job by highlighting the kinome in PMNs.

1. It would have been more insightful if they had included patient-derived PMNs or Tumor associated Neutrophils (TAN).

2. the manuscript can be strengthened by validating the targets from the Kinome.

I feel that this manuscript is only preliminary results and needs more to be accepted for publication.

**Do you want your identity to be public for this peer review?** For information about this choice, including consent withdrawal, please see our Privacy Policy

Reviewer #1: No

Reviewer #2: No

---

## [Author Response · Author response to Decision Letter 1]

5 Feb 2026

Response to Reviewers – PONE-D-25-47347

Point by point reply

Dear Academic Editor and Reviewers,

We would like to thank the Academic Editor and the reviewers for their careful evaluation of our manuscript entitled

“Characterization of the TREM-1 Signaling Landscape in Human Neutrophils” (PONE-D-25-47347).

We appreciate the constructive comments and believe that the revisions have substantially improved the clarity, transparency, and scientific rigor of the manuscript.

Below, we provide a detailed, point-by-point response to all editorial and reviewer comments. All changes have been incorporated into the revised manuscript and are highlighted in the version submitted with track changes.

Editorial Requirements

1. PLOS ONE style requirements

Response:

The manuscript has been reformatted according to the official PLOS ONE templates for the main text, title page, and author affiliations. File naming conventions have also been adjusted to comply with journal requirements.

Manuscript change:

Formatting updated throughout the manuscript.

2. Participant consent

Response:

We have expanded the ethics statement in the Methods section to explicitly state that informed consent was obtained from all participants and that consent was obtained in written form. No minors were included in this study.

Manuscript change:

Methods section, Ethics statement.

3. Role of the funders

Response:

We confirm that the funders had no role in study design, data collection and analysis, decision to publish, or preparation of the manuscript. This statement has been added to the cover letter as requested.

Cover letter addition:

4. Data availability

Response:

We have updated the Data Availability Statement to clarify that all data underlying the findings are now deposited in publicly accessible repositories. Accession numbers and links are provided in the revised manuscript.

Manuscript change:

Data Availability section.

5. Use of “data not shown”

Response:

All instances of the phrase “data not shown” have been removed. Where relevant, corresponding data have either been included in the manuscript, added as Supporting Information, or deposited in a public repository with appropriate references.

Manuscript change:

Results and Discussion sections.

6. Placement of ethics statement

Response:

The ethics statement now appears exclusively in the Methods section and has been removed from all other sections.

Manuscript change:

Methods section.

7. Reviewer-recommended citations

Response:

All suggested references were carefully evaluated and cited where relevant to the revised manuscript.

Reviewer #1

Comment 1: Donor characteristics and sample size

Response:

We agree that donor characteristics are important for reproducibility. We have expanded the Methods section to include age range and sex distribution of donors. We acknowledge the limited number of donors as a limitation inherent to studies using primary human neutrophils and have explicitly addressed this point in the Discussion.

Manuscript change:

Methods (Donors) and Discussion (Limitations).

Comment 2: Typographical inconsistencies in cell numbers

Response:

We thank the reviewer for pointing this out. All reported cell numbers have been carefully reviewed and corrected for consistency and accuracy.

Manuscript change:

Methods section.

Comment 3: Annotation of Figure 1

Response:

Key kinases have now been explicitly labeled in Figure 1 (a–d) to improve readability and highlight relevant signaling components.

Manuscript change:

Figure 1 and corresponding legend.

Comment 4: Overlap between TREM-1 and TLR4 signaling

Response:

We agree that LPS can indirectly amplify TREM-1 signaling. This conceptual limitation is now explicitly discussed. Our intention was not to claim complete separation of TLR4- and TREM-1-mediated pathways, but rather to identify shared and distinct signaling tendencies. We have clarified this rationale and interpretation in the Discussion.

Manuscript change:

Discussion section.

Comment 5: Font size in Figure 3

Response:

Font sizes in Figure 3 (a–d) have been increased to improve readability.

Manuscript change:

Figure 3.

Comment 6: Dissecting TLR4 versus TREM-1 signaling

Response:

We acknowledge that genetic knockdown approaches would be required for definitive pathway separation. However, primary human neutrophils are not amenable to stable genetic manipulation. We have clarified this methodological limitation and framed our conclusions accordingly.

Manuscript change:

Discussion (Limitations).

Comment 7: Lack of functional assessment

Response:

We agree that functional validation would further strengthen the study. The current work was designed as a comprehensive signaling and transcriptomic characterization. We now clearly state this scope and limitation and outline future directions for functional validation.

Manuscript change:

Discussion section.

Reviewer #2

Comment 1: Inclusion of patient-derived PMNs or TANs

Response:

We agree that patient-derived cells would be highly informative. However, this study was intentionally designed to establish a baseline signaling landscape in healthy human neutrophils. This rationale and limitation are now explicitly discussed.

Manuscript change:

Discussion section.

Comment 2: Validation of kinome targets

Response:

We acknowledge this point and now emphasize that the kinome analysis provides a hypothesis-generating framework. Functional validation of selected targets is an important next step and is highlighted as future work.

Manuscript change:

Discussion section.

Concluding remarks

We believe that the revised manuscript addresses all concerns raised by the editor and reviewers and that it now meets the publication criteria of PLOS ONE. We sincerely thank the editor and reviewers for their insightful comments and the opportunity to revise our work.

Kind regards,

Markus P. Radsak

on behalf of all authors

---

## [Decision Letter · Decision Letter 1]

17 Feb 2026

Characterization of the TREM-1 Signaling Landscape in Human Neutrophils

PONE-D-25-47347R1

Dear Dr. Radsak,

We’re pleased to inform you that your manuscript has been judged scientifically suitable for publication and will be formally accepted for publication once it meets all outstanding technical requirements.

Kind regards,

Qi Wu

Academic Editor

PLOS One

Additional Editor Comments (optional):

Reviewers' comments:

Reviewer's Responses to Questions

**Comments to the Author**

Reviewer #1: All comments have been addressed

2. Is the manuscript technically sound, and do the data support the conclusions?

Reviewer #1: Yes

3. Has the statistical analysis been performed appropriately and rigorously?

Reviewer #1: Yes

4. Have the authors made all data underlying the findings in their manuscript fully available?

Reviewer #1: Yes

5. Is the manuscript presented in an intelligible fashion and written in standard English?

Reviewer #1: Yes

Reviewer #1: The research manuscript provides new insight into the signature of TREM1 signaling landscape in human neutrophils.

In the first round of the Revision the authors have done a good job in addressing all me comments and revising the manuscript.

Overall, this is an interesting study that provides valuable new information.

**Do you want your identity to be public for this peer review?** For information about this choice, including consent withdrawal, please see our Privacy Policy

Reviewer #1: No

---

## [Editor Report · Acceptance letter]

PONE-D-25-47347R1

PLOS One

Dear Dr. Radsak,

I'm pleased to inform you that your manuscript has been deemed suitable for publication in PLOS One. Congratulations! Your manuscript is now being handed over to our production team.

Kind regards,

on behalf of

Dr. Qi Wu

Academic Editor

PLOS One